# Phenotypic Heterogeneity of Variably Protease-Sensitive Prionopathy: A Report of Three Cases Carrying Different Genotypes at *PRNP* Codon 129

**DOI:** 10.3390/v14020367

**Published:** 2022-02-10

**Authors:** Simone Baiardi, Angela Mammana, Marcello Rossi, Anna Ladogana, Benedetta Carlà, Pierluigi Gambetti, Sabina Capellari, Piero Parchi

**Affiliations:** 1IRCCS Istituto delle Scienze Neurologiche di Bologna, 40139 Bologna, Italy; simone.baiardi6@unibo.it (S.B.); angela.mammana2@unibo.it (A.M.); marcello.rossi@ausl.bologna.it (M.R.); bens.ca92@gmail.com (B.C.); sabina.capellari@unibo.it (S.C.); 2Department of Experimental, Diagnostic and Specialty Medicine (DIMES), University of Bologna, 40139 Bologna, Italy; 3Department of Neuroscience, Istituto Superiore di Sanità, 00161 Rome, Italy; anna.ladogana@iss.it; 4Department of Pathology, School of Medicine, Case Western Reserve University, Cleveland, OH 44106, USA; pxg13@case.edu; 5Department of Biomedical and Neuromotor Sciences (DIBINEM), University of Bologna, 40139 Bologna, Italy

**Keywords:** VPSPr, prion disease, *PRNP*, protein misfolding, scrapie, amyloid, real-time quaking-induced conversion, RT-QuIC, Creutzfeldt-Jakob disease, CJD

## Abstract

Variably protease-sensitive prionopathy is an exceedingly rare, likely underestimated, sporadic prion disease that is characterized by heterogeneous and often non-specific clinical and pathological features posing diagnostic challenges. We report the results of a comprehensive analysis of three emblematic cases carrying different genotypes at the methionine (M)/valine (V) polymorphic codon 129 in the prion protein gene (*PRNP*). Clinical, biochemical, and neuropathological findings highlighted the prominent role of the host genetic background as a phenotypic modulator. In particular, the *PRNP* codon 129 showed a remarkable influence on the physicochemical properties of the pathological prion protein (PrP^Sc^), especially on the sensitivity to proteinase K (PK) digestion (VV > MV > MM), which variably affected the three main fragments (i.e., of 19, 17, and 7 kDa, respectively) comprising the PrP^Sc^ profile after PK digestion and immunoblotting. This, in turn, correlated with significant differences in the ratio between the 19 kDa and the 7 kDa fragments which was highest in the MM case and lowest in the VV one. The relative amount of cerebral and cerebellar PrP mini-plaques immunohistochemistry showed a similar association with the codon 129 genotype (i.e., VV > MV > MM). Clinical manifestations and results of diagnostic investigations were non-specific, except for the detection of prion seeding activity by the real-time quaking-induced conversion assay in the only cerebrospinal fluid sample that we tested (from patient 129VV).

## 1. Introduction

Prion diseases are a group of rare heterogeneous neurodegenerative disorders of humans and other mammals characterized by tissue deposition of an abnormal isoform (PrP^Sc^) of the cellular prion protein (PrP^C^) enriched in β-sheet secondary structure [1]. Unlike PrP^C^, PrP^Sc^ is partially resistant to protease digestion and highly prone to form aggregates and detergent-insoluble polymers [2]. By acquiring neurotoxic properties, PrP^Sc^ aggregates lead to synaptic loss, microglial activation, spongiform change, astrocytic gliosis, and eventually neuronal loss [3]. Human prion diseases occur most frequently as sporadic disorders of unknown origin [4]. However, pathogenic mutations in the prion protein gene (*PRNP)*, encoding for the cellular prion protein (PrP^C^) [5], or the accidental prion transmission through medical procedures or foodcontaminated with the PrP^Sc^ associated with the bovine spongiform encephalopathy can also cause the disease [4]. Sporadic prion diseases in humans include seven subtypes with distinctive clinicopathological and molecular features [3]. A total of six belong to the sporadic Creutzfeldt-Jakob disease (sCJD) spectrum [6,7], while the seventh most divergent phenotype carries the name variably protease-sensitive prionopathy (VPSPr). While sCJD has an incidence of about two cases per million, to date, VPSPr has only been reported in approximately 40 individuals worldwide [8].

Originally described by Gambetti et al. as protease-sensitive prionopathy in a series of 11 patients carrying valine homozygosity (VV) at the *PNRP* codon 129 [9], VPSPr owes this name to the variable resistance of PrP^Sc^ to the digestion by proteinase K (PK) that is overall significantly lower than in sCJD [10]. VPSPr also shows peculiar histopathologic and clinical features and a distinctive ladder-like Western blot profile of the PK-resistant PrP^Sc^ core fragments [8,11]. The “atypical” clinical phenotype that often does not immediately raise the suspicion of prion disease [12,13,14,15], the frequent inconclusive results of diagnostic tests, and the difficulties in PrP^Sc^ detection due to its sensitivity to PK digestion likely make VPSPr the most underdiagnosed sporadic prion disease.

Aiming to contribute to the definition of the phenotypic spectrum of VPSPr, and increase awareness among clinicians and neuropathologists, here we report in detail three novels, emblematic cases (including the first VPSPr patient ever recognized worldwide) carrying different genotypes at codon 129 of *PRNP* highlighting their effect on the disease-associated phenotype.

## 2. Materials and Methods

### 2.1. Molecular Genetic Analysis

To rule out mutations and define the genotype at the polymorphic codon 129, we sequenced the *PRNP* open reading frame, as previously described [16].

### 2.2. Clinical Evaluation and Diagnostic Tests

All three patients underwent comprehensive neurological evaluations and electroencephalographic recordings (EEG). Brain magnetic resonance imaging (MRI) was available in Cases #2 and #3, and cerebrospinal fluid (CSF) only in the latter. In CSF, the total-tau (t-tau), phosphorylated-tau (p-tau), and amyloid-beta 1-42 and 1-40 (Aβ42 and Aβ40) levels were analyzed using commercially available ELISA kits (INNOTEST htau-Ag, INNOTEST phosphorylated-tau181, and INNOTEST Aβ142 and Aβ1–40, Innogenetics/Fujirebio, Gent, Belgium). The Aβ42/Aβ40 ratio was calculated as previously described [17]. CSF protein 14-3-3 was evaluated semi-quantitatively by comparing the Western Blot signals of the tested sample in comparison to those of the control samples (with a weak or a strong 14-3-3 signal, respectively), as previously described [18]. Finally, CSF prion real-time quaking-induced conversion (RT-QuIC) assays were performed with both full-length (23–31 aa) and truncated (90–231 aa) recombinant Syrian hamster prion protein (rPrP), according to previously published protocols [18,19].

### 2.3. Neuropathological Evaluation

For neuropathological and biochemical analyses, the brain tissue was obtained post-mortem according to a standard protocol [20]. Tissue from the right hemisphere was immediately frozen at −80 °C while the left hemisphere was fixed in 10% formalin and processed by paraffin wax embedding. Semi-quantitative evaluation of spongiform change, gliosis, and neuronal loss was carried out in 22 brain regions by comparing hematoxylin and eosin-stained sections. For PrP immunohistochemistry, paraffin sections from formalin-fixed and formic acid-treated blocks of the selected brain regions were processed using the monoclonal antibody 3F4 (1:400, Signet Labs, Dedham, MA, USA). Moreover, we performed immunohistochemistry with antibodies that were specific for the activated microglia (anti-HLA, clone CR3/43, dilution 1:100, Agilent Dako, Santa Clara, CA, USA), and astroglia (anti-GFAP, clone 6F2, dilution 1:100, Agilent Dako), Aβ (4G8, dilution 1:5000, Signet Labs), p-tau (AT8, dilution 1:200, Innogenetics), transactive response DNA binding protein 43 kDa (TDP-43) (anti-phospho TDP-43 pS409/410, dilution 1:5000, CosmoBio, Carlsbad, CA, USA), and α-synuclein (LB509, dilution 1:100, Thermo Fisher Scientific, Waltham, MA, USA), using several brain regions, mainly following established consensus criteria [21,22,23,24].

### 2.4. Biochemical Analysis

#### 2.4.1. Brain Homogenate Processing

Brain homogenates (TH, 10% *w/v*) of cortical gray matter, striatum, thalamus, and cerebellum were prepared either in a lysis buffer (LB, 100 mM Tris, 100 mM NaCl, 10 mM EDTA, 0.5% Nonidet P-40, 0.5% sodium deoxycholate, pH 7.4) or in a buffer containing sarkosyl, as previously reported [10]. All the experiments were done using the samples in LB except for the analysis of P3 fraction that was obtained after PrP^Sc^ purification in sarkosyl [10].

#### 2.4.2. PrP Deglycosylation

N-linked glycans were removed using a peptide-N-glycosidase F (PNGase F) kit (New England Biolabs, Ipswich, MA, USA) according to the manufacturer’s instructions.

#### 2.4.3. PK Titration Curves

Grey matter TH were digested using serial dilutions of PK activity ranging from 0.125 to 4 U/mL for Case #1 and Case #2, and from 0.0625 to 1 U/mL for Case #3. The samples were incubated 1 h at 37 °C under mild shaking (300 rpm).

#### 2.4.4. Western Blot

Western blots were performed as described [25]. The samples were run in a 7 cm long separating gel and then transferred to Immobilon-P membranes (Millipore). The monoclonal antibodies 3F4 (epitope at PrP residues 108–111, 1:30,000 working dilution), T2 (epitope at PrP residues 132-217, 1:5000 working dilution) [26], and an N-terminal polyclonal antibody (epitope at PrP residues 23–40, 1:2000 working dilution) [27] were used as primary probes. The immunoreactive signal was visualized by enhanced chemiluminescence (Immobilon Western, Merck Millipore, Burlington, MA, USA) on a LAS 3000 camera (Fujifilm, Tokio, Japan).

## 3. Results

### 3.1. Molecular Genetic Analysis

Genetic analysis ruled out pathogenic *PRNP* mutations and revealed the codon 129 genotypes methionine/methionine (MM), methionine/valine (MV), and valine/valine (VV) in Cases #1 to #3, respectively.

### 3.2. Clinical Features

Case #1. A 67-year-old man was referred to a Movement Disorders Clinic for suspicion of multiple system atrophy. His past medical history was unremarkable. About six months before admission, he complained of dizziness, progressive walking difficulties, and memory and writing troubles shortly afterward. At the first neurological examination, he was alert, bradyphrenic, apraxic, and oriented in space but not in time. The Mini-Mental state examination score was 23/30 (normal values > 24). A motor exam revealed symmetric dystonic posture of upper limb extremities, dysdiadochokinesia, action tremor, and paratonia. Deep tendon reflexes were brisk, and a Babinski sign was evident on the left. Gait was ataxic with decreased arm swings. Over the next year, he developed cortical sensory dysfunction and myoclonus late in the disease course. He died 30 months after the disease onset. EEG recordings never showed either periodic or epileptic discharges.

Case #2. A 79-year-old man presented with short-term memory loss and apathy, at first ascribed to mood disorder. In the following months, the cognitive status rapidly deteriorated until a loss of independence in daily activities led to the suspicion of Alzheimer’s disease (AD). His clinical condition abruptly worsened after a convulsive seizure that was triggered by fever, and concurrent hypertensive crisis: the patient became unresponsive and neurological examination revealed right-sided head deviation, ipsilateral hypertonia, primitive reflexes, diffuse brisk deep tendon reflexes, and bilateral Babinski sign. At this time, an EEG showed a generalized slowing of the background activity, while a brain MRI revealed non-specific hyperintensities of periventricular white matter on T2 sequences. The patient died six months after the first symptom appearance.

Case #3. A 72-year-old woman presented with behavioral change (i.e., diminished social interest, apathy) and memory loss. After three months, at the first neurological examination, the patient appeared disoriented and remarkably worried. She also showed involuntary and repetitive slow movements of the left foot, dysmetria, ataxic gait, increased deep tendon reflexes, and a Babinski sign. A detailed neuropsychological evaluation disclosed a severe attention deficit, memory deficiency mainly in tests exploring the recall abilities, anosognosia, constructive apraxia, and expressive aphasia. At this time, EEG disclosed a normal background activity with frontal intermittent rhythmic delta activity, and a brain MRI showed non-specific white matter hyperintensities on T2 sequences, whereas diffusion-weighted imaging was unremarkable. CSF analyses disclosed positivity for protein 14-3-3, increased levels of t-tau (1273 pg/mL) and p-tau (140 pg/mL), reduced Aβ42/Aβ40 ratio (0.44), and positive prion seeding activity by RT-QuIC (i.e., only using Syrian hamster rPrP 90–231). The patient slowly continued to progress until akinetic mutism and died three years after onset.

A summary of the main clinical and laboratory findings in the three patients is shown in Table 1.

### 3.3. Neuropathology

Case #1. The macroscopic evaluation showed diffuse cerebellar atrophy with prominent involvement of the cerebellar vermis. A histological examination revealed mild to moderate spongiform change, neuronal loss, and gliosis in all cerebral cortices, cerebellum, striatum, and thalamus, whereas the brainstem and hypothalamus were virtually spared. The entorhinal cortex showed only focal spongiform change, whereas the CA1 region of the hippocampus and the subiculum appeared unaffected. Overall, these histopathological changes were mild despite the relatively long clinical course (2.5 years). Notably, the spongiform change, which consisted of a mixture of fine and intermediate vacuoles (the latter smaller than those observed in sCJDMM2) [9], was significantly more evident in the occipital cortex, in the molecular layer of the cerebellum (Figure 1A,B), and, focally, in the medial thalamus. Marked astrogliosis and microglial activation were detected in the neocortices and subcortical white matter of all cerebral lobes (Figure 1C–E). At immunohistochemistry, the dominant pattern of PrP deposition was represented by small plaque-like deposits, which were most evident in the molecular layer of the cerebellum, thalamus, hippocampus, and the deep layer of the cerebral cortices (Figure 1F–I). In addition, fine, granular PrP deposits that focally surrounded the vacuoles were noted, especially in the neocortices. Rare tau-positive neurons, glial cells, and scattered neuropil threads were detected in the trans-entorhinal and entorhinal cortices. Finally, immunostainings for Aβ, α-synuclein, and TDP-43 were negative.

Case #2. This patient presented the shortest disease duration and, overall, the mildest neuropathological changes. Microscopic analysis revealed focal spongiform change that was characterized by fine vacuoles of intermediate size throughout neocortices, where it was slightly more pronounced in the occipital cortex, and in the striatum (Figure 2A). Spongiform change was not evident in the thalamus, limbic structures, brainstem, and cerebellum (Figure 2B). All the anatomic areas that are reported above, but the cerebellum and brainstem, showed moderate to severe neuronal loss and astrocytic and microglial activation (Figure 2C–E). Immunohistochemistry for PrP disclosed a dot-like deposition pattern and, to a lesser extent, small plaque-like, often surrounded by dot/granule clusters in all the cerebral cortices, thalamus, entorhinal cortex, and basal forebrain (Figure 2F–H). No PrP-immunoreactivity was detected in the cerebellum. Immunochemistry for Aβ revealed diffuse deposits and rare core plaques throughout the cerebral cortices, the CA1 sector of the hippocampus, and the striatum. Few sparse foci were observed in the midbrain (Thal phase 4) [21] (Figure 2I). Cerebral amyloid angiopathy (CAA) was detected in the meningeal vessels (Figure 2K). Rare tau-positive neurofibrillary tangles were present in the transentorhinal cortex (Braak tau-stage I) [28]. No α-synuclein and TDP-43 immunoreactivity was detected.

Case #3. Microscopic examination revealed mild spongiform change comprising both small and intermediate vacuoles in the cerebral cortex of all lobes, the molecular layer of the cerebellum (Figure 3A,B), and the striatum. Remarkably, gliosis and neuronal loss were more pronounced than the spongiform change in both the cortical and subcortical structures (basal ganglia, diencephalon, limbic system, and brainstem) (Figure 3C). Microglial activation was also severe in the white matter (Figure 3D). Immunohistochemistry revealed PrP mini-plaques and granular deposits, often organized in clusters, in the molecular layer of the cerebellum (Figure 3E,F). A synaptic or a dot-like staining pattern was visible in the thalamus and hippocampus. (Figure 3G). Aβ deposits in diffuse plaques were evident in the cerebral cortices and hippocampus (Figure 3H), while there were rare core plaques only in the frontal cortex (Thal phase 2) [21]. CAA was also noted. There were tau-positive neurofibrillary tangles and neuropil threads in the entorhinal cortex, CA1 sector, subiculum, and temporo-occipital gyrus (Braak tau-stage III) [28] (Figure 3I). Moreover, focal and widespread fine granular deposits into the astrocytic processes were detected in both the gray and white matter with subpial, subependymal, and perivascular distribution, consistently with aging-related tau astrogliopathy (ARTAG) (Figure 3K) [29]. Immunochemistry for α-synuclein and TDP-43 was negative.

### 3.4. Biochemical PrP^Sc^ Analysis

Western blot analysis of TH from the three VPSPr brains, probed with an antibody against PrP N-terminus, revealed the typical electrophoretic profile of full-length PrP comprising three bands representing the unglycosylated, the monoglycosylated, and the diglycosylated isoforms of the protein (Figure 4A). However, after separation of PrP^C^ from PrP^Sc^ by sarkosyl extraction of insoluble PrP^Sc^, only the unglycosylated and the monoglycosylated fragments remained visible, indicating that in VPSPr diglycosylated PrP^C^ does not convert to PrP^Sc^. Indeed, despite the presence of a normally glycosylated PrP^C^, VPSPr PrP^Sc^ lacks the diglycosylated isoform regardless of the genotype at codon 129 (Figure 4A).

After PK digestion of TH, the samples that were probed with the mAb 3F4 (Figure 4B left panel) or T2 (data not shown) showed a 7 kDa band with the highest intensity in the 129VV case, the lowest in the 129MM, and intermediate in the 129MV. Moreover, additional bands migrating approximatively at 26, 23, 19, and 17 kDa were detected in 129MM and 129MV cases, representing mono- and unglycosylated PrP^Sc^ fragments of 19 and 17 kDa, as best appreciated after glycan removal (Figure 4B right panel). Interestingly, the 129VV case showed a slightly different pattern comprising faint 26 nd 19 kDa bands and two additional fragments migrating slightly faster than 23 kDa and 17 kDa. The latter finding was also confirmed after PNGase treatment. Deglycosylation best revealed the distinctive “quantitative” profiles of unglycosylated fragments among the three cases: the 19 kDa band predominated over the 17 kDa peptide in the 129MM genotype, conversely the faster migrating 17 kDa band showed higher intensity than the 19 kDa band in the VV genotype; finally, in the 129MV genotype, the two fragments showed a similar intensity.

Although the overall VPSPr-PrP^Sc^ is more sensitive than CJD-PrP^Sc^ (Appendix A) [10], PK titration curves showed significant heterogeneity also among the VPSPr samples. Indeed, PrP^Sc^ resistance was highest in VPSPr 129MM, lowest in the 129VV case, and intermediate in VPSPr 129MV (Figure 5). However, we found that this difference in PK sensitivity mainly concerned the 19 kDa peptide, whereas the 17 kDa and especially the 7 kDa fragments were less sensitive to PK digestion than the 19 kDa peptide.

## 4. Discussion

VPSPr, the last identified human prion disease, still represents a puzzling entity because of its rarity and the significant heterogeneity of its clinical and histo-molecular features. The present work supports the current view that the genotype at codon 129 plays an essential role in determining the phenotypic diversity in VPSPr.

In line with previous studies [11,30], we found that VPSPr PrP^Sc^ shows the highest sensitivity to PK digestion in individuals 129VV, the lowest in those carrying 129MM, and intermediate values in those with 129MV. We also confirmed that VPSPr PrP^Sc^ shows an overall distinctive immunoblot profile, but with an additional significant heterogeneity among cases depending on the PK activity and the codon 129 genotype [31]. Overall, both 3F4 and T2 antibodies detect five PrP^Sc^ fragments independent from the codon 129 genotype, including mono- and un-glycosylated forms of a 19 kDa fragment corresponding to CJD PrP^Sc^ type 2, monoglycosylated and unglycosylated forms of a 17 kDa fragment lacking the GPI anchor and a 7 kDa fragment ragged at both N- and C-termini.

Our comparison of the PrP immunoblot profile between the TH and a P3 fraction of sarkosyl-insoluble full-length PrP^Sc^ demonstrated that all three PrP^C^ glycoforms are normally expressed in VPSPr brains and that the lack of the diglycosylated isoform is a specific feature of VPSPr PrP^Sc^ related to the lack of conversion of diglycosylated PrP^C^ into PrP^Sc^. The finding further supports the current view that strain-specific PrP^Sc^ structural constraints may determine the selective recruitment of PrP^Sc^ glycoform favoring those with reduced or no glycan chains, shifting the ratios of glycoforms within PrP^Sc^ toward mono- and un-glycosylated glycoforms [32].

Of note, we found that the relative proportion of the three unglycosylated PrP^Sc^ fragments is significantly associated with the codon 129 genotype. The relatively high representation and high PK resistance of the 19 kDa band in the 129MM brain make this case the most “CJD-like” among the three. On the opposite side, the PrP^Sc^ that is associated with 129VV showed the most atypical profile, characterized by the rapid disappearance of PrP^Sc^ type 2 due to its high PK sensitivity, a slightly faster migration of the 17 kDa fragment, and the highest intensity of the 7 kDa band. It is well established that similar PrP^Sc^ fragments also characterize the so-called inherited prion protein amyloidoses, including Gerstmann-Sträussler-Scheinker disease (GSS), and the cerebral and systemic amyloidosis that are linked to stop-codon truncating *PRNP* mutations [33]. Although PrP^Sc^ does not accumulate as amyloid in VPSPr, it is noteworthy that the amount of PrP mini-plaques that were seen at immunochemistry has been linked to the relative amount of the 7 kDa fragment (VV > MV > MM) [11] [present work]. Further similarities between GSS and VPSPr are the relatively long disease duration and the lack of efficient transmission to transgenic mice expressing human PrP [34,35].

In VPSPr, the different PrP^Sc^ biochemical properties influence the clinicopathological features, including the disease duration, spongiform change severity, and PrP deposition pattern at immunohistochemistry. Conversely, the type of spongiform change, characterized by small and intermediate vacuoles and the lesion distribution profile, appear consistent across codon 129 genotypes. The short duration that is associated with very mild spongiform change and the sparing of the cerebellum in our 129MV case are unlikely related to the natural history of the disease, but rather depend on the early occurrence of medical complications, such as fever and hypertensive crisis, that lead to the early death of the patient.

In our case series, the clinical presentations were heterogeneous, largely non-specific, and included a variable combination of cognitive, psychiatric, and cerebellar signs. Prion disease (i.e., CJD) was only suspected in the 129VV case, whereas other alternative diagnoses were AD in the 129MV case and atypical multiple system atrophy in the patient carrying 129MM. Both the EEG and brain MRI disclosed non-specific and inconclusive findings. CSF analyses were performed only in the 129VV case: the positivity of 14-3-3 protein raised the suspicion of CJD, but the concurrently increased levels of p-tau and decreased Aβ ratio prompted the diagnosis of AD. In this context, the detection of prion seeding activity by CSF RT-QuIC steered the diagnosis in the direction of prion disorder. Despite the limited number of cases that were analyzed, CSF RT-QuIC seems to have a good performance in VPSPr, being positive in five out of six patients (83.3%) that were tested using the truncated rPrP [19,36]. Future studies should confirm this initial evidence in larger cohorts, since the specificity of prion RT-QuIC makes the assay a suitable diagnostic tool to identify novel VPSPr cases clinically.

We also investigated VPSPr brains for neurodegenerative co-pathologies. We detected CAA and a low burden of AD neuropathologic change [37] in the older patients (129MV and VV). Various degrees of Aβ-immunoreactivity have been previously reported in VPSPr and attributed to aging [9,11]. Co-existence of widespread neocortical Lewy bodies, mild AD neuropathologic change, and CAA have previously been reported in a single VPSPr-129MV patient [38]. Besides tau neurofibrillary pathology, we detected ARTAG pathology in the 129VV case. Prominent gray matter ARTAG was previously reported in sporadic and genetic CJD [39], but not in VPSPr. Further studies are needed to confirm the consistency of this association.

Finally, Case#1 of our series deserves a particular comment being the first case of the disease that was later identified and named VPSPr, which was recognized as an atypical novel human prion disease phenotype that was not classifiable within the sCJD spectrum [40].

## 5. Conclusions

The present study strengthens previously published evidence that, in VPSPr, the *PRNP* codon 129 has a remarkable influence on pathologic and biochemical features, and, to a lesser extent, on clinical presentation. This variability makes both clinical and neuropathologic diagnosis challenging. On the one hand, neuropathologists should raise the suspicion of VPSPr when intermediate sized vacuoles and PrP mini-plaques or small plaque-like deposits are identified, even in front of an apparently “negative” result of PrP^Sc^ typing by Western blot. On the other hand, neurologists should consider VPSPr in the differential diagnosis with other forms of dementia that are associated with motor/behavioral symptoms, in particular frontotemporal dementia, and atypical forms of AD. The detection of CSF prion seeding activity by RT-QuIC in the single VPSPr patient that we tested (129VV) is consistent with the few other reports that are available in the literature, suggesting that RT-QuIC might be a useful tool for the identification of this rare and puzzling disease in vivo.

## Figures and Tables

**Figure 1 viruses-14-00367-f001:**
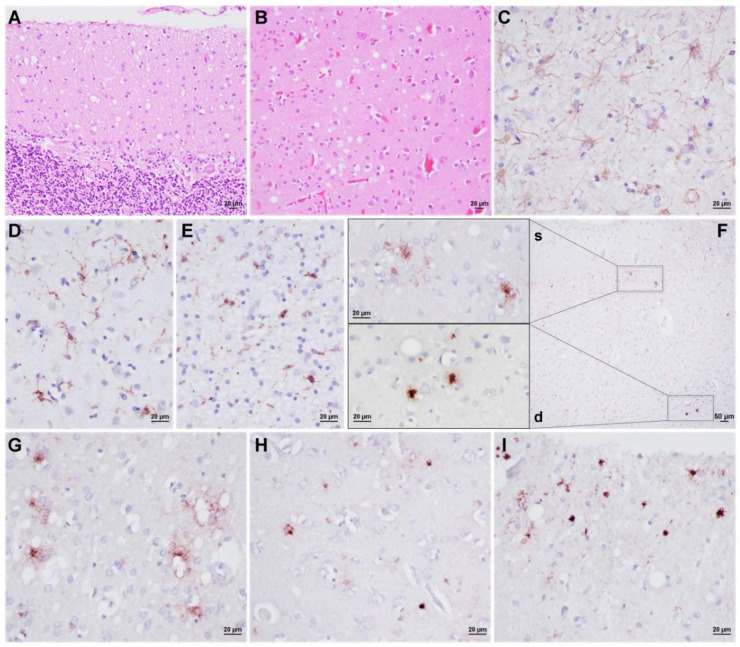
Neuropathological features of VPSPr 129MM (Case#1). (**A**,**B**) Spongiform change with vacuoles of “intermediate” size in the cerebellum molecular layer and occipital cortex. (**C**) Activated astrocytes in the occipital cortex. (**D**,**E**) Marked microglial activation involving both the occipital gray and white matter. (**F**) PrP deposits in the occipital cortex: on the left, the upper left box shows delicate, patchy granular PrP deposits in the superficial cortical layers, while the lower box small plaque-like PrP deposits in the deep layers. (**G**) Granular PrP deposits surrounding the vacuoles (occipital cortex). (**H**,**I**) Small plaque-like PrP deposits in the entorhinal cortex and cerebellar molecular layer. Haematoxylin-eosin staining (**A**,**B**), and immunochemistry for GFAP (**C**), HLA (**D**,**E**), and PrP (**F**–**I**). Legend: s, superficial; d, deep.

**Figure 2 viruses-14-00367-f002:**
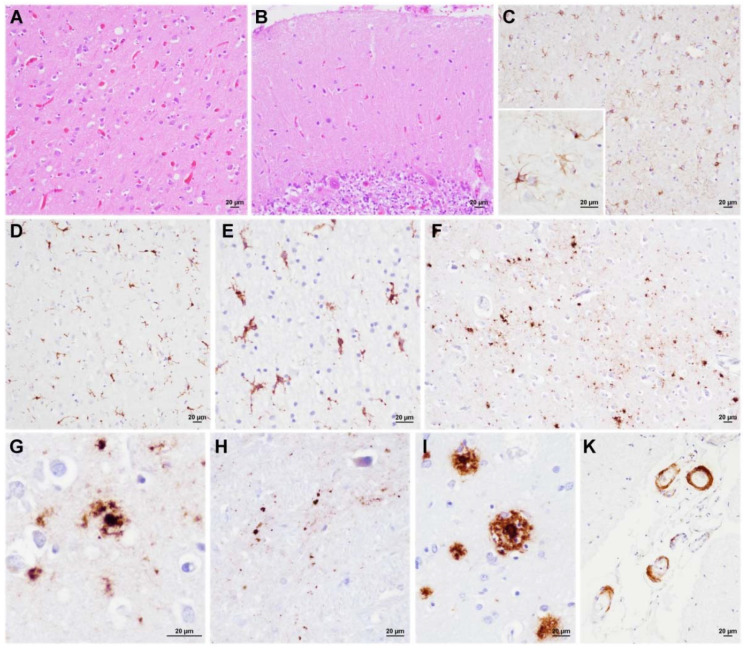
Neuropathological features of VPSPr 129MV (Case#2). (**A**) Mild spongiform change in the occipital cortex. (**B**) Lack of spongiform change in the cerebellum. (**C**–**E**) Severe astrocytic (the lower box shows a detail at higher magnification) and microglial activation in the parietal gray (**C**,**D**) and white matter (**E**). (**F**–**H**) PrP dot-like and mini-plaque deposits in the occipital cortex (**F**,**G**) and thalamus (**H**). (**G**) PrP mini-plaque that was surrounded by a cluster of fine granules. (**I**) Aβ core plaque in the frontal cortex. (**K**) Cerebral Aβ angiopathy in the meningeal vessels of occipital lobe. Haematoxylin-eosin staining (**A**,**B**), and immunochemistry for GFAP (**C**), HLA (**D**,**E**), PrP (**F**–**H**), and Aβ (**I**,**K**).

**Figure 3 viruses-14-00367-f003:**
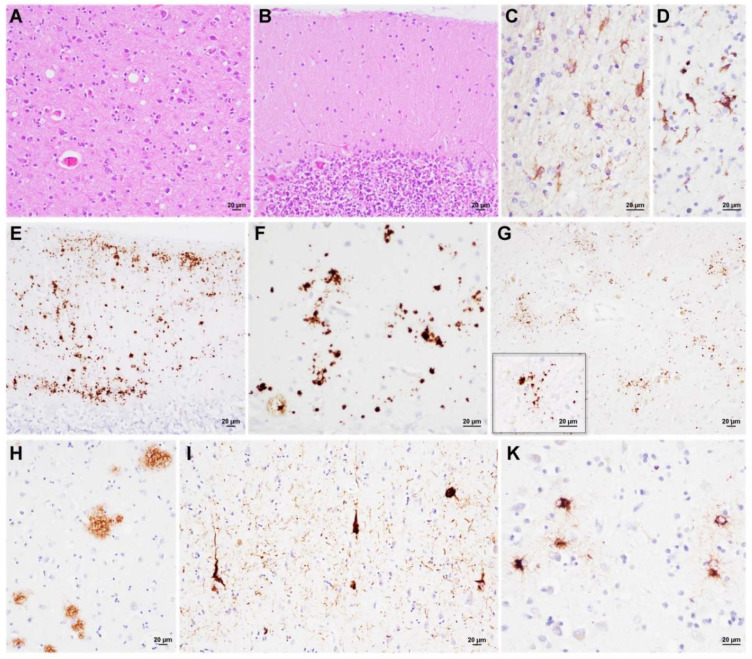
Neuropathological features of VPSPr 129VV (Case#3). (**A**,**B**) Mild spongiform change that was characterized by a mixture of small and intermediate size vacuoles in the occipital cortex and cerebellum. (**C**,**D**) Astro- and microglial activation in the temporal gray and occipital white matter, respectively. (**E**,**F**) PrP granular and mini-plaque deposits in the occipital cortex. (**G**) Clusters of dot-like PrP deposits in the medial thalamus. On the left, the lower box shows a detail at higher magnification. (**H**) Diffuse Aβ deposits in the frontal cortex. (**I**) Tau-positive globular and flame-shape neurofibrillary tangles and neuropil threads in the entorhinal cortex. (**K**) Fuzzy tau-immunoreactive astrocytes in the occipital cortex. Haematoxylin-eosin staining (**A**,**B**), and immunochemistry for GFAP (**C**), HLA (**D**), PrP (**E**–**G**), Aβ (**H**), and p-tau (**I**,**K**).

**Figure 4 viruses-14-00367-f004:**
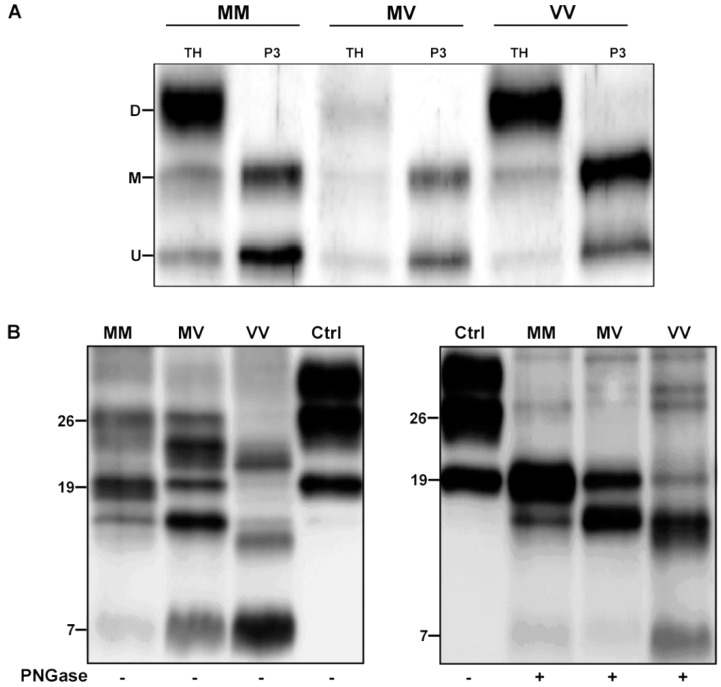
Immunoblotting of proteinase K (PK)-untreated and -treated PrP from VPSPr affected brains. (**A**) Comparison of the electrophoretic profile of full-length PK-untreated PrP in total homogenate (TH) and purified (P3) samples from the VPSPr cases carrying 129VV, 129 MM, and 129 MV. The membrane was probed with the polyclonal PrP 23-40 antibody. (**B**) The electrophoretic profiles of proteinase K-treated TH from VPSPr cases before (left) and following PNGase treatment (right), which revealed the different pattern of unglycosylated fragments among the three cases. The membranes were probed with mAb 3F4. A sCJD VV2 case was included as a control. Approximate molecular masses are in kilodaltons. M, methionine; V, valine; D, diglycosylated; M, monoglycosylated; U, unglycosylated; Ctrl, control.

**Figure 5 viruses-14-00367-f005:**
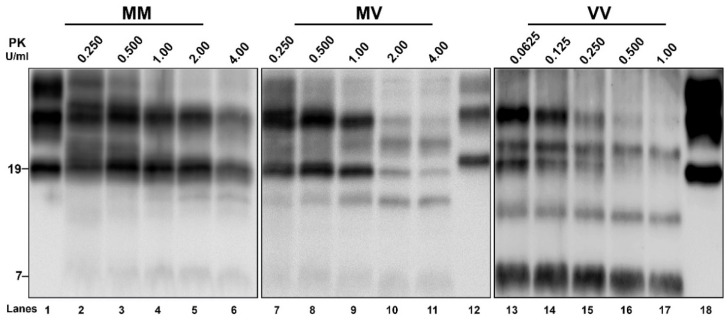
Analysis of PrP^Sc^ protease-resistance by PK titration. Brain homogenates from 129VV, 129MM, and 129MV cases were run after digestion with increasing amounts of PK to highlight the variable PK resistance of the PrP^Sc^ fragments that are associated with the three codon 129 genotypes. In the 129VV case, we used lower PK concentrations than in 129MM and 129MV because of the higher sensitivity of PrP^Sc^ to PK digestion, as demonstrated by the more significant signal loss in the three points of the curves with identical PK levels (i.e., 0.250 U/mL, 0.500 U/mL, 1.00 U/mL). Note the striking difference in the degree of PK resistance of the 19 kDa fragment among the three cases. The membranes were probed with mAb 3F4. Approximate molecular masses are in kilodaltons. Lane 1 sCJD VV2; Lane 12 sCJD MM1; Lane 18 sCJD VV2. M, methionine; V, valine.

**Table 1 viruses-14-00367-t001:** Main clinical findings in VPSPr cases.

Case	Codon 129 Genotype	Age at Onset	Disease Duration	Presenting Features	EEG	MRI	CSF 14-3-3	CSF RT-QuIC	Alternative Diagnosis
#1	Met/Met	67	2.5	Gait unsteadiness	Non-specific	Na	Na	Na	MSA
#2	Met/Val	79	0.5	Memory loss, behavioral changes	Non-specific	Non-specific T2 white matter hyperintensities	Na	Na	AD
#3	Val/Val	72	3.0	Memory loss, behavioral changes	Non-specific	Non-specific T2- white matter hyperintensities	+	+	CJD

Age at onset and disease duration are expressed in years. List of abbreviations: Met, methionine; Val, valine; Na, not available; +, positive; MSA, multiple system atrophy; RT-QuIC, real-time quaking-induced conversion; AD, Alzheimer’s disease; CJD, Creutzfeldt-Jakob disease.

## Data Availability

Data supporting reported results are available on reasonable request from the corresponding author.

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
