# Peer review of "Phenotypic Heterogeneity of Variably Protease-Sensitive Prionopathy: A Report of Three Cases Carrying Different Genotypes at PRNP Codon 129"

_viruses, 2022, doi:10.3390/v14020367_

Round 1
Reviewer 1 Report
Recently Variably protease-sensitive prionopathy (VPSPr) was reported and VPSPr is an exceedingly rare. In VPSPr cases, the PRNP codon 129 showed a remarkable influence on the three unglycosylated PrPSc fragments detected by immunoblotting. The ratio between the 8 kDa fragment with ragged N- and C- termini and the 19 kDa fragment was VV>MV>MM. T The relative amount of cerebral and cerebellar PrP mini-plaques at immuno-histochemistry showed a similar association with the codon 129 genotype. Clinical manifestations and results of diagnostic investigations were mainly nonspecific
This manuscript is the most important and interesting work on prion diseases in the last two years, and I spent several days reading the entire paper.
It is a really important study, but the paper has only three cases and the title, abstract and conclusion are overly conclusions based on a hypothesis with very little evidence.
Major points:
- In abstract, #3 has a detailed CSF examination, but the other cases do not have detailed CSF examination. The conclusion of the abstract identified that the detection of CSF prion seeding activity by the real-time quaking-induced conversion assay in the single CSF sample analyzed supports current evidence indicating such a test as the most promising for diagnosing VPSPr in vivo. If all cases were positive by RT-QUIC, this conclusion would be acceptable, but there is insufficient evidence for this conclusion when only one case was examined.
- In the figure 4 and PrPSc fragments detected by immunoblotting, our team has focused on 8kDa and19kDa and studied it in a considerable number of cases. However, the amount of 8kDa and 19kDa fragments varies considerably even in MM1 cases. It is too unclear whether it makes sense to take the ratio of that 8 kDa to 19 kDa fragment, and the number of papers cited is small.
Author Response
Point-by-point response to Reviewer 1.
- In abstract, #3 has a detailed CSF examination, but the other cases do not have detailed CSF examination. The abstract's conclusion identified that the detection of CSF prion seeding activity by the real-time quaking-induced conversion assay in the single CSF sample analyzed supports current evidence indicating such a test as the most promising for diagnosing VPSPr in vivo. If all cases were positive by RT-QUIC, this conclusion would be acceptable, but there is insufficient evidence for this conclusion when only one case was examined.
Response. We changed the title, the last sentence of the abstract, and the conclusion section accordingly. Specifically, we toned down the emphasis on the role of CSF RT-QuIC in the clinical diagnosis (abstract and conclusion) and of codon 129 genotype as phenotypic modulator in VPSPr (title).
- In the figure 4 and PrPSc fragments detected by immunoblotting, our team has focused on 8kDa and 19kDa and studied it in a considerable number of cases. However, the amount of 8kDa and 19kDa fragments varies considerably even in MM1 cases. It is too unclear whether it makes sense to take the ratio of that 8 kDa to 19 kDa fragment, and the number of papers cited is small.
Response. The formation of a significant amount of unglycosylated PrPSc fragments in the 7-9 kDa range characterizes VPSPr and GSS, while is not seen in CJD or FFI. Therefore, we found unclear the reviewer’s statement about the MM1 cases, the most common CJD subtype. Nevertheless, we modified the text in the abstract to better explain our observation on the ratio between the two fragments and put less emphasis on it. Finally, we believe we have cited all the most significant articles presenting data on PrPSc typing in VPSPr.
Reviewer 2 Report
In this paper Baiardi et. al. attempt to characterize the effect genotype (MM/MV/VV; condon 129 of the PRNP gene) has on clinical features, neuropathology, and biochemical analysis on PrPsc in a rare variably protease-resistant prionopathy (VPSPr) of three patients, each of which carry a different polymorphism. The manuscript is well written. While the description of the clinical features and neuropathology were thorough and complete, the biochemical PrPsc analysis section was difficult to follow as the progression of experiments was unclear.
Major Point:
- In the biochemical PrPsc analysis section, tissue homogenates (TH) were first compared pre and post sarkosyl separation by western blot. This is followed by PK digestion but the sample is not specified – is that the TH or the P3 fraction? Likewise, it was unclear if PNGase was used on samples that had been PK treated or not. Appears that PNGase was used post-PK treatment based on the figure legend? Please clarify.
- For the PK titration curves, why were 2 separate antibodies used when only 3F4 was used in the previous experiment? Additionally, there appears to be discrepancies in the banding patterns of the MV and VV compared to the PK digested samples in Fig. 4B. Is there an explanation for that? Could you also explain how the signal intensity of the MM blot appears to increase as the PK concentration increases? Is the blot reversed? And finally, if the controls are sCJD VV2 and VPSPr’s are more sensitive and distinguishable by PK treatment, it would be helpful to see a PK titration on the sCJD VV2 (or all 3 genotypes) as a comparison/control.
- CSF RT-QuIC analysis – Only one CSF sample was assayed but the discussion states 5 of 6 VPSPr samples were positive in QuIC. I agree that RT-QuIC is one of the better diagnostic tools for early detection and for this paper, steering an unknown diagnosis. Your statements though in both discussion and conclusion paragraphs present this option last but with a heavy emphasis on the assay being key to identifying these types of cases. Perhaps rework if this is what you are trying to say.
Minor Points:
- Abstract states an 8kDa fraction repeatedly but all the results in biochemical analysis states a 7 kDa fraction.
- Methods, line 105: Use of (TH) – please use consistently throughout the manuscript. Referred to as brain, tissue, and total homogenate.
- Methods, line113: PK use for case #1 & #2 states 0.125 to 2 U/ml. Figure 5 has a range of 0.125 to 4 U/ml
- Line 225: Second word in most I think should be more?
Author Response
Point-by-point response to Reviewer 2.
In the biochemical PrPsc analysis section, tissue homogenates (TH) were first compared pre and post-sarkosyl separation by western blot. This is followed by PK digestion but the sample is not specified – is that the TH or the P3 fraction? Likewise, it was unclear if PNGase was used on samples that had been PK treated or not. Appears that PNGase was used post-PK treatment based on the figure legend? Please clarify.
Response. PK digestion was performed on TH. The P3 fraction was only used to demonstrate the lack of diglycosylated PrPSc before PK treatment. PNGase was used post-PK treatment. We have now specified this detail in the method section (page 3, lines 111-115) and in the figure legend.
For the PK titration curves, why were 2 separate antibodies used when only 3F4 was used in the previous experiment?
Response. We included in the figure the WB stained with the T2 antibody (MM case) because we thought that the 7 kDa band was well seen without overexposing the image. However, we have now found an acceptable WB stained with 3F4 and replaced the one stained with T2 with this one.
Additionally, there appears to be discrepancies in the banding patterns of the MV and VV compared to the PK digested samples in Fig. 4B. Is there an explanation for that?
Response. The explanation relies on the different degrees of PK resistance of the three main fragments comprising the WB PrPSc profile in VPSPr MV and VV. Indeed, the PK-titration curve nicely shows that the ratio between these bands changes according to the PK activity used for digestion.
Could you also explain how the signal intensity of the MM blot appears to increase as the PK concentration increases? Is the blot reversed?
Response. A limited variation of intensity of the signal in the range of PK activity having a similar effect on PrPSc digestion may be due to an imperfect manual loading of the sample in the different wells. Anyway, after changing the image (see above), the signal does not appear to increase anymore.
And finally, if the controls are sCJD VV2 and VPSPr’s are more sensitive and distinguishable by PK treatment, it would be helpful to see a PK titration on the sCJD VV2 (or all 3 genotypes) as a comparison/control.
Response. We added in the supplementary material an example of PK titration on the sCJDVV2 TH, showing the striking difference with the VPSPr VV sample.
CSF RT-QuIC analysis – Only one CSF sample was assayed but the discussion states 5 of 6 VPSPr samples were positive in QuIC. I agree that RT-QuIC is one of the better diagnostic tools for early detection and for this paper, steering an unknown diagnosis. Your statements though in both discussion and conclusion paragraphs present this option last but with a heavy emphasis on the assay being key to identifying these types of cases. Perhaps rework if this is what you are trying to say.
Response. We changed the last sentence of the abstract, and the conclusion section accordingly (see also the response to reviewer 1).
Minor Points:
- Abstract states an 8kDa fraction repeatedly but all the results in biochemical analysis states a 7 kDa fraction.
We corrected the discrepancy in the abstract.
- Methods, line 105: Use of (TH) – please use consistently throughout the manuscript. Referred to as brain, tissue, and total homogenate.
We have now used the term TH consistently to refer to total brain homogenates.
- Methods, line113: PK use for case #1 & #2 states 0.125 to 2 U/ml. Figure 5 has a range of 0.125 to 4 U/ml
We corrected the discrepancy in the method section.
- Line 225: Second word in most I think should be more?
We substituted most with more.
Round 2
Reviewer 1 Report
Thank for revise of the manuscript. I've revised the manuscript and improved it considerably. 
#1.In abstract, the authors removed the sentence ;except for the detection of prion seeding activity by the real-time quaking-induced conversion assay in the single CSF sample analyzed (129VV). Analysis using the RT-QUIC method was attempted in only one of the three cases. This may give wrong information to the readers
#2.in figure 5, the condition of PK digestion in MM and VV was different. the author should match the PK requirements of all lanes. The discussion should be based on that assumption.
#3. False positives occur in about 1% of RT-QUIC assays, making RT-QUIC a questionable assay for specificity
#4.The word "CSF prion RT-QuIC" feels too unnatural in the last sentence.  Please change it to another word.
Author Response
REVIEWER 1
Thank for revise of the manuscript. I've revised the manuscript and improved it considerably.
#1.In abstract, the authors removed the sentence except for the detection of prion seeding activity by the real-time quaking-induced conversion assay in the single CSF sample analyzed (129VV). Analysis using the RT-QUIC method was attempted in only one of the three cases. This may give wrong information to the readers
RESPONSE: According to the Reviewer's suggestion, we specified further that we tested by RT-QuIC only the CSF of the patient carrying 129VV.
#2.in figure 5, the condition of PK digestion in MM and VV was different. the author should match the PK requirements of all lanes. The discussion should be based on that assumption.
We fully understand the Reviewer's comment, but we prefer not to modify the figure given that the primary goal of the image was not the detailed comparison of the effect of a wide range of PK activities but rather show the variability of signal loss among the PrPSc fragments in the three cases. (i.e., the change in the ratio among the fragments, especially the 19 and 7 kDa bands associated with increasing PK concentrations in the three cases). PrPSc in VPSPr 129VV is much more sensitive than in VPSPr 129MM and 129MV, as demonstrated by the three points of the curves shared by all cases (i.e., 0.250 U/ml, 0.500 U/ml, 1 U/ml). Therefore, using higher PK concentrations in the 129VV case would only provide a complete signal loss, a result that goes beyond the primary aim we thought for this figure. Nonetheless, following the Reviewer's comment, we have provided a better explanation for the different ranges of PK activity used in the three cases in the figure legend.
#3. False positives occur in about 1% of RT-QUIC assays, making RT-QUIC a questionable assay for specificity
RESPONSE: We think that 99% specificity is quite good for a diagnostic test.
#4.The word "CSF prion RT-QuIC" feels too unnatural in the last sentence. Please change it to another word.
RESPONSE: We slightly modified the sentence accordingly, as suggested.
Reviewer 2 Report
Thank you for addressing previous comments.
I have only 3 minor formatting issues.
- Line 29 abstract - extra space and period
- Move figure 4 down one line as 268-268 are separated
- Move figure 5 down as it is in the middle of the paragraph and separated from the figure legend.
Author Response
REVIEWER 2
Thank you for addressing previous comments.
I have only 3 minor formatting issues.
Line 29 abstract - extra space and period
Move figure 4 down one line as 268-268 are separated
Move figure 5 down as it is in the middle of the paragraph and separated from the figure legend.
We re-formatted the manuscript according to the Reviewer’s suggestions.